# Control Efficiency and Yield Response of Chemical and Biological Treatments against Fruit Rot of Arecanut: A Network Meta-Analysis

**DOI:** 10.3390/jof8090937

**Published:** 2022-09-05

**Authors:** Balanagouda Patil, Shankarappa Sridhara, Hanumappa Narayanaswamy, Vinayaka Hegde, Ajay Kumar Mishra

**Affiliations:** 1Department of Plant Pathology, University of Agricultural and Horticultural Sciences, Shivamogga 577255, Karnataka, India; 2Division of Crop Protection, ICAR-Central Plantation Crops Research Institute, Kasaragod 671124, Kerala, India; 3Center for Climate Resilient Agriculture, University of Agricultural and Horticultural Sciences, Shivamogga 577255, Karnataka, India; 4Khalifa Centre for Genetic Engineering and Biotechnology, United Arab Emirates University, Al Ain P.O. Box 15551, United Arab Emirates

**Keywords:** *Areca catechu*, fruit rot disease, network meta-analysis, control efficiency, yield response

## Abstract

Fruit rot disease (FRD) in arecanut has appeared in most of the arecanut growing regions of India in the last few decades. A few comprehensive studies on the management of FRD under field conditions have examined various treatment combinations for disease control and yield response analysis. This study aimed to compare the control efficiencies and yield responses of treatments applied over multiple locations and compute the probable returns of investment (ROIs) for treatment costs. Data were gathered from 21 field trials conducted across five main arecanut growing regions of India in the period 2012–2019. The collected data were subjected to analysis with a multivariate (network) meta-analytical model, following standard statistical protocols. The quantitative, synthesized data were evaluated for the estimated effects of disease pressure (DP_Low_ ≤ 35% of FRD^Inc^ in the treatments > DP_High_), mean disease control efficiencies (treatment mean, *C*), and yield responses (*R*) corresponding to the tested treatments. Based on disease control efficacy, the evaluated treatments were grouped into three efficacy groups (EGs): higher EGs were observed for the Bordeaux mixture (*C*, 81.94%) and its stabilized formulation (*C*, 74.99%), Metalaxyl + Mancozeb (*C*, 70.66%), while lower EGs were observed in plots treated with Biofight (*C*, 29.91%), Biopot (*C*, 25.66%), and Suraksha (*C*, 29.74%) and intermediate EGs were observed in plots to which microbial consortia (bio-agents) had been applied. Disease pressure acted as a significant moderator variable, influencing yield response and gain. At DP_Low_, the Bordeaux fungicide mixture (102%, 22% of increased yield) and Metalaxyl + Mancozeb (77.5%, +15.5%) exhibited higher yield responses, with absolute arecanut yield gains of 916.5 kg ha^−1^ and 884 kg ha^−1^, while, under DP_High,_ Fosetyl-AL (819.6 kg ha^−1^) showed a yield response of 90.5%. To ensure maximum yield sustainability, arecanut growers should focus on the spraying of fungicides (a mixture of different active ingredients or formulations or products) as a preventative measure, followed by treating palms with either soil microbial consortia or commercial formulations of organic fungicides.

## 1. Introduction

Fruit rot disease (FRD), caused by *Phytophthora meadii* McRae [1], is the most devastating and destructive arecanut (*Areca catechu* L.) disease, causing huge economic losses to growers and putting the greatest constraint on arecanut production [2,3]. FRD was first reported in arecanut in India in 1906 [4], and further concurrent occurrences of the disease were identified in most of the arecanut growing regions of India [5]. Since its first report in India, FRD has been observed as a disease of less importance and scope. However, in recent years, FRD incidence has increased due to the build-up of inoculum, the growing of susceptible varieties, monotony in crop management strategies, prevailing conducive environmental factors, and reduced base-line sensitivity due to general use of chemical fungicides against pathogens [6,7,8,9,10]. FRD has now become endemic in various arecanut-cultivating agro-climatic regions of India, including neighboring countries where arecanut is widely grown.

The symptoms of FRD are mainly recognized as rotting and extensive shedding of immature nuts, which lie near the base of the palms [4]. Initially, dark-green water-soaked lesions appear closer to the perianth region of the nut surface and spread gradually, covering the entire nut [5,11]. Under hot–humid conditions, infected nuts are enveloped with a white mycelial mat and severe infection may reach fruit stalks, including inflorescence, leading to stalk or inflorescence rot [12,13,14]. Heavy infection with pathogens might lead to quantitative and qualitative losses of areca nuts, and infected nuts are not suitable for chewing or masticatory purposes. At a domestic level, considerable yield losses of areca nuts due to FRD frequently occur in endemic areas—losses greater than those sustained on newly established (non-traditional areas) arecanut plantations that coincide with prevailing environmental conditions [15]. The occurrence of FRD primarily depends on patterns of rainfall (RF, in mm), intra- and inter-plot relative humidity (RH, in %), and temperature (T, in °C). Under such conditions, fungicidal sprays need to be resorted to in order to curtail the impacts of the disease as well as reduce yield losses to the growers. Due to the lack of sources of resistance/tolerance to FRD, fungicide applications have been considered the sole options for managing the disease effectively under field conditions and achieving good yield responses [5,16,17].

Even though various control measures have been recommended over the years, considerable economic losses due to FRD are quite usual. For two decades, there has been significantly increased interest in evaluating fungicidal efficacy against FRD. Many researchers (in the public and private domains) across the country have made efforts over the years to bring out effective fungicides and bio-agents [2]. To date, a considerable number of field trials have compared the effects of fungicidal sprays, including yield responses, and data have been gathered from online published articles, including scientific reports from research institutes across the country. Through a network of field trials involving uniform or different treatments, the control efficiencies and yield responses for currently available and labeled fungicides against FRD have been evaluated in the arecanut growing regions of India [18,19,20,21,22,23].

However, in most of the previous studies on the efficacy of fungicides and yield responses, only means (standard errors of the means) and treatment effects (statistically significant and critical differences) have been reported. This may not provide sufficient knowledge to support farmers in their decision-making regarding disease management strategies. With the aims of doubling farmers’ incomes and minimizing the inessential or needless harmful effects of fungicidal sprays, there must be an appraisal of economic cost–benefit analyses [24,25,26]. An approach which considered multiple locations and years would be more likely to generate information relevant to the estimation of the profitable returns of fungicide applications. This information could help growers develop and decide whether to use fungicides or not, and even help them to decide on the most suitable fungicides for control of FRD under different conditions [25,27].

To the best of our knowledge, there is a lack of comprehensive studies on the impacts of fungicide applications (single or multi-site in nature) in arecanut cultivation across varied environments and production strategies with profitable returns. Such comprehensive analyses aid in drawing informative conclusions on fungicide efficacy and yield response with profitability in arecanut growing. Available studies lack models or quantification of the effects of moderator variables on the control efficiencies and yield responses of evaluated fungicides. However, meta-analysis is a suggestive approach for the comparison, integration, and interpretation of results from individual experiments [28,29].

Owing to the complexity of analyzing datasets from multi-trials across the country, meta-analysis has appeared as a suitable approach in plant pathology for dealing with large, complex, and multi-site, year-round datasets [24,27,30,31]. This tool was developed for social science studies, and it is now considered a basic computational tool for plant pathologists who analyze big, heterogeneous, multi-site analyses, particularly when computing fungicide efficiency and performing profit returns analyses over differed environmental profiles [26,32,33]. A meta-analysis is a quantitative synthesis and narrative review of research findings that provides an estimation of the overall effect size of the variables under study and of random effects and between-study variance [28,32]. This sort of analysis allows the determination of the magnitude and significance of treatment effects in relation to control efficiency and profitable returns along with the impacts of moderator variables on fungicide efficacy.

This study aimed to perform a quantitative, synthetic analysis of the efficacy of fungicides and biological treatments against arecanut FRD tested in arecanut growing regions of India and to estimate the FRD control efficiencies and yield responses of the treatments.

## 2. Materials and Methods

### 2.1. Dataset, Experimental Design, and Application

All data utilized in this study were synthesized from 6 reports (published in reputed national and international journals) containing results from small plot or on-farm experiments. A total of 21 field trials conducted during six growing seasons (monsoon, June–September months) from 2012 to 2019 years across the main Indian arecanut growing regions were included in the analysis (Figure 1). Fungicide trials were carried out in well-established arecanut gardens with FRD on susceptible varieties, and the field trials were managed by following the recommended package of practices (PoP).

Protection treatments consisted of the application (3 sprays) of labeled chemical fungicides or biological products against arecanut FRD (registered under the Ministry of Agriculture and Farmers Welfare, India), with 3–5 repetitions under the field conditions, as detailed in Table 1 and Table 2. The tested fungicides belonged to the inorganic (IO), phenylalanine (PAL), phenyl amide (PA), dithiocarbamate (DCR), phosphonate (PSN), and carboxylic acid (CAA) groups, having single or multi-site action against the pathogen. Bio-agents and commercially available enriched natural products (Nutri-rich plant hormone formulations) were evaluated along with labeled fungicides against FRD for effective management. The bunch-spraying system was set up in the monsoon seasons of 2014 to 2019 with the aforementioned fungicides with single and multi-site actions, including bio-agents and Nutri-rich products. The spraying operations were carried out with a crown gold portable rocket sprayer with a heavy delivery set (5 m long) with a nut nipple connection, which was directed to treat the bunch zone effectively. A pump barrel and pressure chamber were installed to make the sprayer convenient to operate by hand, and the spray lance was fitted with a triple-action brass nozzle which could apply fungicides up to a height of 20–30 ft.

All of the trials were carried out with various treatments, which were completely randomized in different blocks and replicated 3–5 times, following a uniform protocol, including a non-fungicide-treated control or a reference against which remaining treatments were compared. Spacing between rows and palms in the gardens was 2.7 m (9 ft), and 50 palms per treatment were considered for the application of treatments, with the remaining untreated palms being changed every year to avoid extensive development of the disease over the years/seasons. The spraying operations were performed thrice in a season, the first before the onset of the monsoon (the last week of May) as a preventative measure and the second as well as the third 30–45 days after the first or second spray (the first fortnight of July) for the curative approach. Similarly, bio-control agents were applied thrice in a season near the effective root zones, which were 1 m apart. Palms were sprayed with 5 L of each fungicide with adhesive, i.e., RENTON VA 98,055 (or water for control), per plot, and phytosanitation was kept constant for all treatments.

### 2.2. Meta-Analytical Synthesis and Effect Size

In network meta-analysis, effect sizes are statistics that consist of means and their ratios and the variation between treatments and controls, which can be utilized to assess the comprehensive effects of treatments, including the strength of correlations between variables [26,34]. To estimate a comprehensive size effect, the random effects of meta-analytical approaches use different sources of variation that are quite similar in multi-location trials by considering the weightage of each experiment, which is an inverse function of the within-study and between-study variance [26,32].

### 2.3. Disease Control Analysis

The naturally log-transformed rate ratio of FRD incidence (*L^inc^*) was calculated for each fungicidal treatment as a measure of efficiency [24], as expressed in Equation (1):
(1)$$L^{inc}=\ln\;\left(\frac{Inc_{Trt}}{Inc_{Check}}\right)=\ln\;(Inc_{Trt})\;-\;\ln\;(Inc_{Check})$$
where *Inc_Trt_* is the average FRD incidence of treated plots and *Inc_check_* represents the mean FRD incidence for the untreated control (check). As the difference between log means equals the log ratio, the right-hand side of the equation is the equivalent form of the log response ratio. *L^Inc^* (derived after fitting the meta-analytical model) was back-transformed to provide estimates of control efficiency (%) for easier comprehension of the results [24] and calculated using the following Equation (2):
(2)$$C\;=\;(1\;-\;\exp\;(L^{Inc}))\;\times \;100$$

A large, negative *L^Inc^* value, by default, relates to a large, positive *C*, higher relative efficiency. Only trials in which the untreated plots with FRD incidences greater than 25% were kept for the analysis of control efficacy.

### 2.4. Yield Response Analysis

As a measure of yield response for the evaluated treatments, two effect sizes were estimated: the absolute yield difference (*D*, calculated as the difference between each fungicide treatment’s mean and the untreated control) and the relative yield response (*L^yld^*) [26,27,30]. The latter values were calculated using an equation with the same form as Equation 1, then *L^yld^* was back-converted to generate yield response estimates as percentages (*R*, %), using Equation (3) [24]:
(3)$$L^{yld}=\ln\;\left(\frac{Yld_{Trt}}{Yld_{Check}}\right)=\ln\;(Yld_{Trt})\;-\;\ln\;(Yld_{Check})\;$$
where *Yld_Trt_* indicates the mean arecanut yield of fungicide treatment and *Yld_Check_* is the average arecanut yield of the untreated check [24]. Then, the percent yield response (*R*) was computed from *L^yld^* using the following Equation (4):
(4)$$R\;=\;(\exp\;(L^{yld})\;-\;1)\;\times \;100$$
where *L^yld^* represents the log-transformed mean response rate of the yield as estimated through the meta-analytical approach for fungicide treatments and *R* indicates relative yield response as a percentage.

### 2.5. Quantitative Data Synthesis

Multi-treatment or network meta-analysis techniques were used to determine *L^inc^*, *L^yld^*, and *D*, because field trials with different combinations of treatments were evaluated under multi-environment situations. In such conditions, researchers often choose a separate univariate meta-analysis approach for each size effect of interest (for each combination of treatment mean effect); however, by employing this method, there is a chance of missing the relationships between effect size estimates within trials, which could produce biased results of estimates [24,32]. Hence, the multivariate meta-analytical tool allowed us to compare the combined effect of treatments across the studies, since the treatment of Bordeaux mixture (1%) was present in all the trials. In the network meta-analysis, two kinds of effect sizes were estimated by considering the differences between the treatments with fungicides and the untreated controls, which is commonly called the conditional modeling approach. It is a simple approach that was performed in this study to fit a two-way linear mixed model by taking into account treatment means from individual trials in a two-stage analysis [29].

To measure log response rates, the meta-analytical model was fitted to log-transformed estimated means, while, to measure *D*, the model was embedded directly in the absolute mean arecanut yield data. All measures of modeling were based on the estimation of within-study variances, and the weights of the means and the log means were as described in Paul et al. [24,30]. In short, in all cases, the untreated control was used as a reference, and the standard model is represented in Equation (5):
(5)$$Y_{i}\sim N\;(µ,\;\sum +S_{i})$$
where Y_i_ is the vector of responses in the ith trial and is thought to have a normal distribution, with mean effect size µ, and Si represents a variance–covariance matrix of random effects of the ith trial, where Σ is the 7 × 7 between-trial variance–covariance matrix [35]. An informal Σ matrix was used, and data models were fitted with parameters deemed most suitable according to the maximum-likelihood method. The R_METAFOR_ package [36] was used to fit all meta-analytical models. The Wald statistical test was employed to assess the consistency or inconsistency between and among the trials, including the influence of moderator variables on the outcomes of treatments.

Due to the significance of the study, the effects of FRD pressure (*DP*) and the year in which a field experiment was conducted were evaluated as moderator variables. To analyze fungicide efficacy effects, the individual field trials were classified as low FRD pressure (*DP*, based on each corresponding untreated check mean): *DP*_Low_ ≤ 35% or *DP*_High_ > 35%. Similarly, we classified each trial into two yield level (*YL*) categories based on the BM1% mean yield for each trial: *YL*_Low_ ≤ 1.25 kg/palm or *YL*_High_ > 1.25 kg/palm, for the analysis of yield variables. Finally, the experimental year was assessed as a moderator variable to determine whether there were any dynamic changes in the control efficiency trend for the Bordeaux mixture (present in all treatment sets) over the whole period covered in this quantitative review. All R codes, data, and plots have been made available at https://github.com/juanchiem/arecanut_rot_meta for reproducibility.

## 3. Results

### 3.1. Epidemics and Yields in Treated/Control Plots

The field trials were carried out in areas where fruit rot disease (FRD) was endemic and had considerable prevalence (availability of initial inoculum). FRD occurrence was determined in all of the evaluated trial locations. Disease incidence in untreated control plots ranged from 20.5 to 92.7%, with an average incidence of 42.25% (Figure 2).

There was significant variation (*p* ≤ 0.05) in FRD incidence between and within the years considered, which was reflected in site-specific differences in disease incidence and prevailing conditions of climate, topography, and soil profiles. As expected, the average level of incidence was lower in fungicide-treated plots compared to untreated checks (Figure 2). Among the evaluated treatments, including fungicides, bio-agents, and nutrient mixtures, the fungicide-treated plots showed reduced disease incidence (*DI* ≤ 35%), followed by bio-agent consortia, which had median incidence values of 35–48.5%. A relatively higher disease incidence was observed in the remaining plots treated with Biofight, Biopot, and Suraksha (*DI* > 45%) compared with the other treatments.

Similar to epidemic levels, there were statistically significant differences (*p* ≤ 0.05) in yield (kg/palm) responses in treated and untreated control plots between the seasons and within (locations) seasons under the multi-environment situations (Figure 2). The baseline yield response ranged from 0.72 to 3.25 kg per palm, with a median value of 1.66 kg per palm across the 21 trials conducted under varied climatic and topographic profiles. As expected, yield levels were higher in fungicide-treated plots than in the untreated checks. Regarding the tested treatments, the highest average yields were observed in fungicide-treated plots (*YL*, 2.50 kg/palm), followed by bio-agent consortia-treated gardens (*YL*, 2.25 kg/palm), while reduced yield levels were noticed in plots treated with Biofight, Biopot, and Suraksha (median value of *YL*, 1.50 kg/palm) (Figure 2).

Based on FRD epidemic levels (disease pressure, *DP*) and baseline yield responses (yield level, *YL*) in the treated plots and untreated controls, all 21 field trials were categorized into low and high disease pressure and yield levels (Figure 3). Low disease pressure was represented as *DP*_Low_ (disease incidence ≤ 35%) and high disease pressure was represented as *DP*_High_ (disease incidence > 35%). The disease incidence in *DP*_Low_ trials ranged from 14.70 to 34.80% (median value of 24.75%), while in *DP*_High_ trials it ranged from 34.80 to 93.50% (median = 64.15%). Similar to disease pressure, 21 field trials were grouped into higher and lower yield levels (*YL*s) and higher-yield-level trials ranged from 2.15 to 2.85 kg per palm, with a median yield of 2.50 kg per palm, whereas the lower-yield-level trails had yield responses of 1.25 to 2.15 kg per palm (median *YL* = 1.70 kg/palm) (Figure 3).

### 3.2. Meta-Analysis of Disease Control Efficiency

#### 3.2.1. High Disease Pressure (*DP*_High_ > 35%)

For all of the treatments that had high disease pressures (*DP*_High_ > 35%), *L*^Inc^ differed substantially from zero, based on the standard normal parameters of the meta-analytical model (*p* = 0.001 for all the treatments). The calculated *L*^Inc^ values ranged from −1.711 to −0.296, with control efficiencies (*C*) of 81.94% and 25.66% for the plots treated with Bordeaux mixture and Biopot, respectively (Table 3). Based on the estimated percentage of control efficiency (*C*), the copper-based fungicides, Bordeaux mixture (81.94%), the stabilized formulation (74.99%), and copper oxychloride (60.87%), and the phenyl amid fungicide metalaxyl + Mancozeb (70.76%) were found to be most effective against FRD in arecanut, while Cymoxanil + Mancozeb (54.25%) had intermediate efficacy against the FRD control and the remaining evaluated nutrient mixtures, such as Biofight (29.91%), Suraksha (29.74%), and Biopot (25.66%), showed the lowest control efficiencies (Figure 4).

The evaluated moderator variables had statistically significant influences on treatment control efficiency (*p* ≤ 0.0001), with residual heterogeneity of 7689.7531 and a moderator coefficient of 358.1598 at a probability (*p*) value ≤ 0.0001. Thus, considering the effects of disease incidence and the years of study as grouped and continuous moderator variables led to a considerable decrease in the between-study variability (data not shown). Based on the results of the Wald parametric test, there was a consistent effect observed in the demonstrated network model (a statistically significant design-by-treatment interaction was observed, *p* ≤ 0.0001).

#### 3.2.2. Low Disease Pressure (*DP*_Low_ ≤ 35%)

The treatments that had lower values of disease pressure (*DP*_Low_ ≤ 35%) and *L^Inc^* values varied statistically from zero, based on the statistical parametric test of the multivariate meta-analysis (*p* ≤ 0.0001, for all the respective treatments evaluated). The estimated levels of *L^Inc^* values ranged from –1.051 to –0.257, with respective disease control efficacies (*C*) ranging from 65.05% for Fenamidone + Mancozeb to 22.69% for the *Trichoderma harzianum* microbial consortium (Table 4). The calculated levels of percentage control efficiency (*C*), generated from naturally logarithmic transformed estimated differences between the treatments and untreated controls, varied from 22.69 to 65.05% (Figure 4). The fungicides, Bordeaux mixture, and Fenamidone + Mancozeb reduced FRD incidence by about 65%, and their effects differed significantly (*p* ≤ 0.0001). These were followed by *Bacillus megatarium*, Blue Bordo at different concentrations, and Bordeaux mixture at a higher concentration than recommended, with control efficiencies varying between 30 and 40%, these differing statistically significantly at *p* ≤ 0.0001, while the lowest efficacy was estimated for the microbial consortia of *Pseudomonas fluorescence* and *Trichoderma harzianum*, their effects differing substantially from those of the other treatments (*p* ≤ 0.0001). The difference between the most and least effective treatments was 42.36 percent of disease control. The Wald statistical test revealed that the network was highly influenced by the design of the experiments (*p* ≤ 0.0001). The effect of moderator variables was comparatively higher among the treatments between trials and within trials (*p* ≤ 0.0001), with residual heterogeneity of 7739.0720 and a moderator co-efficient of 432.0348 at *p* ≤ 0.0001 (Table 4).

### 3.3. Meta-Analysis of Yield Response or Gain

#### 3.3.1. Lower Yield Level (*DP*_High_ > 35%)

The variance for average yields was about 18.5%, taking into account only the untreated control means, and the variance determined when all of the treatments were included (*n* = 58 inputs) was 16.6%. This indicated relatively lower variation in baseline values for yields among field trials and suggested that *D* (level of average difference) could be valuable information as an effect size for assessing the influence of treatments on yield response. The mean yield difference (*D*) values ranged from 0.0857 to 0.646 kg per palm (104 to 839.8 kg ha^−1^), with a median value of 0.371 kg per palm (481 kg ha^−1^). All of the treatments had the lowest mean yield difference values < 0.1 (21.05% of the total inputs, depending on the treatment values shown in Table 5).

The overall calculated *D* values for all the treatments differed statistically significantly from 0, as determined by the standard normal test (*Z*) of the network meta-analytical model (*p* ≤ 0.0001). By way of explanation, the application of fungicides significantly increased the yield responses compared to the untreated checks. For all of the treatments with high disease pressures (*DP*_High_ > 35%), the maximum estimated *D* values were found in plots treated with Bordeaux mixture (0.48 kg/palm, 624 kg ha^−1^) and Fosetyl-AL briquettes with different substrates (0.63 kg/palm (819.6 kg ha^−1^), 0.64 kg/palm (839.8 kg ha^−1^), 0.56 kg/palm (728.3 kg ha^−1^)), while the lowest mean yield differences (*D*) were observed for the application of microbial consortia (bio-agents such as *Bacillus megatarium, Pseudomonas fluorescence,* and *Trichoderma harzianum*), which had a median value of 0.11 kg per palm (143 kg ha^−1^) (Table 5). Based on the Wald statistical parametric test, a lack of inconsistency was observed in the demonstrated network (a statistically non-significant design-by-treatment interaction was found, *p* > 0.05).

#### 3.3.2. Higher Yield Level (*DP*_Low_ ≤ 35%)

The average yield differences (*D*) for all the treatments showed higher yields with relatively lower disease pressures (corresponding to the untreated controls), as estimated by the network meta-analysis, and ranged from 0.02 (26 kg ha^−1^) to 0.705 (916.5 kg ha^−1^). Relatively higher yield gains were estimated for the Bordeaux mixture-treated plots (0.705; 916.5 kg ha^−1^), which differed statistically significantly over all the evaluated treatments (*p* ≤ 0.0001). Further, this treatment was followed by Metalaxyl + Mancozeb (0.533; 692.9 kg ha^−1^), stabilized Bordeaux mixture (0.505; 656.5 kg ha^−1^), and different concentrations of Bordeaux mixture (0.571; 742.3 kg ha^−1^ and 0.490; 637 kg ha^−1^). The latter substantially differed from all the treatments tested at the probability value of *p* ≤ 0.0001 (Table 6). The variance between the estimated averages for the best and least efficient treatments was 0.682 (884 kg ha^−1^). The Wald statistical test for design-by-treatment interactions revealed that the network meta-analysis was consistent (*p* > 0.05).

### 3.4. Effects of Moderator Variables on Yield Responses to Treatments

The influence of the moderator variable ‘disease pressure’ (*DP*), with an FRD incidence threshold limit of 35% (*DP*_Low_ ≤ 35% FRD _Inc_ > *DP*_High_), was statistically significant for *D* (*p* ≤ 0.0001) and *L^Inc^* (*p* ≤ 0.0001). The maximum yield responses (*R*) at *DP*_Low_ were recorded for plots treated with Bordeaux mixture (102.4%, with yield mean difference, *D* = 916.5 kg ha^−1^), with a 22.8% yield increase compared to the untreated controls (Figure 5), while the treatments consisting of applications of Metalaxyl + Mancozeb, copper oxychloride, and other formulations of Bordeaux mixture had yield responses that varied from 58.8 to 77.1%, with an average yield increase of about 12.8% compared to the other treatments. The lowest values for *D* and *R* were estimated for nutrient mixtures, viz., Biofight, Biopot, and Suraksha, which varied from –2.0 to 15% of yield response, with lower yield increases compared to the controls.

Meanwhile, at DP_High_, the highest *R* values were observed for Fosetyl AL briquettes with different substrates, which ranged from 76.5 to 90.8%, corresponding to a 15.6 to 21.7% yield increase relative to the untreated controls. The treatments of Fenamidone + Mancozeb and microbial consortia of bio-agents had marginal yield responses (*R*), which varied from 21.8 to 38.9%, with a 9.25% higher yield of compared to the controls (Figure 5). For some of the treatments, the yield response was not affected by *DP* but substantially influenced the increase in yield (gain) compared to the controls. The relationship between disease control efficiency (*C*) and relative yield response or gain (*R*) was summarized; both classes of disease pressure estimate, *C* and *R*, had significant positive correlations at *DP*_High_ and *DP*_Low_.

## 4. Discussion

Recently, FRD in arecanut started appearing across arecanut growing regions of India and now that it is a major concern it has become the subject of debate. Since the first report of the disease in 1906 [4,5], many fungicides with a single/multi-site mode of action have been evaluated under field conditions as means of minimizing the yield losses sustained by arecanut growers [18,19,20,21,22,23]. In the current study, we utilized a multivariate (network) meta-analytical model to compare the efficacy of integrated treatments, including fungicides, microbial consortia (bio-agents), and nutrient mixtures (growth inducers), evaluated across 21 field trials under a varied range of environments. Previously, simple or univariate meta-analyses have been utilized to summarize the effects of fungicides (univariate) only on disease control (incidence or severity) and yield response for other crop diseases [37]. We further demonstrated the means and uncertainties (effect sizes) of meta-analytical model estimates in disease control and performed yield analyses to inform the decision making of growers for the selection of the most suitable and sustainable management approaches.

As per the previous research on various crops around the world, the network meta-analytical approach permitted us to compare treatments and provide quantitative estimates of effect sizes and their risks by using an available single modeling framework [24,25]. The analysis of raw data published in annual reports and technical bulletins along with unpublished data does not allow for the comparison of treatment means within and between studies, as it fails to generate quantitative estimates of yield responses (differences or gains in yield). The comparison of treatment means and yield responses is key information needed to make a decision based on the disease control efficiencies and yield responses of treatments, as determined by multi-location trials. The network meta-analytical model represented the best approach, since such models provide information with higher statistical accuracy, weight estimates of effect sizes, and can treat the results of different studies as random effects, thereby allowing valuable and conclusive inferences to be drawn [29].

Considerable variation was observed in the efficacy of the integrated treatments tested against FRD, with efficacy ranging from very low for nutrient mixtures (Biofight, Biopot, and Suraksha) to moderate efficacy for the microbial consortia (*Pseudomonas fluorescence*, *Trichoderma harzianum,* and *Bacillus megatarium*) to higher efficiencies for fungicides (copper-based fungicides, such as Bordeaux mixture and copper oxychloride; phosphonates (potassium phosphonate and fosetyl-AL); and phenyl amid and other tested groups of fungicides). Similarly, there was statistically significant variation observed among the integrated treatments, but fungicide-treated plots showed maximal disease control efficiencies compared to the plots that received other treatments. These results confirmed the findings reported in other studies, especially those on soybean target spots around the world [26,38]. Similarly, Belufi et al. [39] reported the highest control efficiency for target spots by sequential application of fungicides, with maximum profitability and economic yield.

Outcomes from the field trials carried out over 21 locations in India to control FRD in arecanut determined that the application of Bordeaux mixture (1%) resulted in a greater reduction in disease incidence compared to the other treatments. These results were corroborated by the report that prophylactic and curative spraying with copper-based fungicides showed promising results in the control of FRD [21,23], which indicated the consistent efficacy of treatments over the trials across arecanut growing regions of India. For instance, an efficacy of treatment 85–90% higher than the corresponding untreated check was observed. However, we concluded that three appropriate timed applications are optimal for better management of FRD and the sustention of yield response.

Although more practical studies are required to gain better understanding of how to control FRD in arecanut under field conditions, some basic concepts in FRD management could be designed based on the data provided by the current study. The major factor that determines the effects of fungicides on yields might be the varieties grown. Due to the lack of availability of varieties of arecanut tolerant of or resistant to FRD, fungicidal efficiency has been greatly influenced and yield response to treatments has varied significantly. On the other hand, the cultivation of varieties susceptible or with low levels of tolerance to FRD combined with management through the application of chemical fungicides seems to be warranted. It is well known that arecanut research conducted by various institutes, universities, and developmental organizations should focus on breeding resistance to FRD-causing pathogens through traditional methods of resistance breeding. Arecanut growers should be more careful while transporting and planting varieties with a history of disease, and nearby fields in which disease is present should be planted with alternate host crops, such as colocasia [40].

The results of the present study were carefully interpreted, as continuous applications of fungicides and other treatments is not recommended [26]. This is because it might lead to the development of fungicide-resistant isolates of pathogens. However, the results of this study are to be used as a guide for assessing and comparing the disease control efficiency ranges of potential fungicides and their influences in terms of yield response as part of the best approach for disease management. Therefore, it would be appropriate to conduct follow-up experiments to test different fungicidal programs, such as different formulations and combi-products, with varying modes of action, spray durations, and numbers of sprays. Reducing the number of sprays by one would lead to a reduction in fungicide application investment and thus increase the recovery of application costs and investments by maximizing yield response. It is also important to prioritize the study of pesticide–crop interactions as well as weather conditions that favor epidemics of FRD in arecanut.

The valuable information generated in our study could be of greater value if it were integrated into a decision-support system that considers not only economic scenarios (arecanut prices and spraying investments) but also disease epidemics [41]. The current database of evaluated trials might provide a valuable resource for the validation of currently available models or the evaluation of different new models specific to Indian conditions in arecanut growing regions.

## 5. Conclusions

In this investigation, fungicides were found to be effective in reducing FRD incidence, and in many cases they significantly increased arecanut yield in arecanut growing regions in India. Despite the variation in the results due to the different fungicides applied and the different management strategies implemented, the outcomes generated by the network meta-analytical model revealed that fungicides remain a prime component of management programs for arecanut FRD. Using the data generated through the network meta-analysis, we demonstrated that three timely applications of copper-based or systemic oomycete-specific fungicides as curative and preventative measures could be reliable means of reducing FRD epidemics. To ensure the maximum yield potential, growers should focus on the spraying of fungicides (mixtures of different active ingredients or formulations or products) as a preventative measure, followed by treating the palms with either soil microbial consortia or defense-inducing organic fungicides. The information generated in this study on the disease control efficiencies and yield responses of various treatment combinations can be used by growers to select more appropriate FRD management programs. Additionally, it would be helpful to develop FRD management approaches to achieve maximum profitability, rates of returns (ROI), and economic benefits by reducing investment in treatments.

## Figures and Tables

**Figure 1 jof-08-00937-f001:**
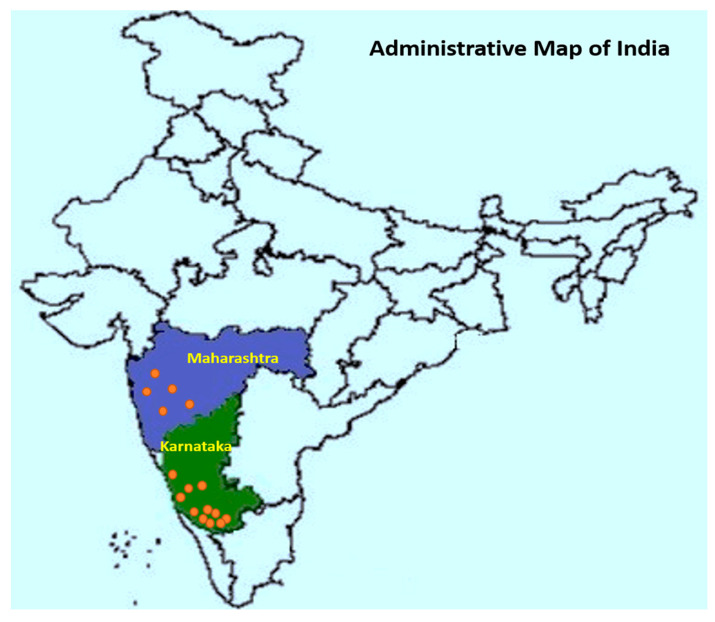
Location map of field trials conducted during the six growing seasons from 2012 to 2019 across the main Indian arecanut growing regions.

**Figure 2 jof-08-00937-f002:**
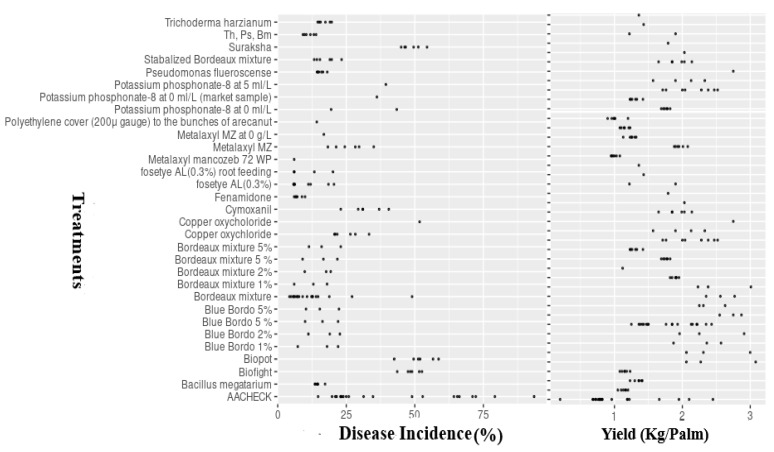
Treatment mean values (points) of FRD incidence and arecanut yield (Kg/Palm). AACHECK = untreated control, with treatments tested at different concentrations. Th = *Trichoderma harzianum*; Ps = *Pseudomonas fluorescence*; Bm = *Bacillus megatarium*.

**Figure 3 jof-08-00937-f003:**
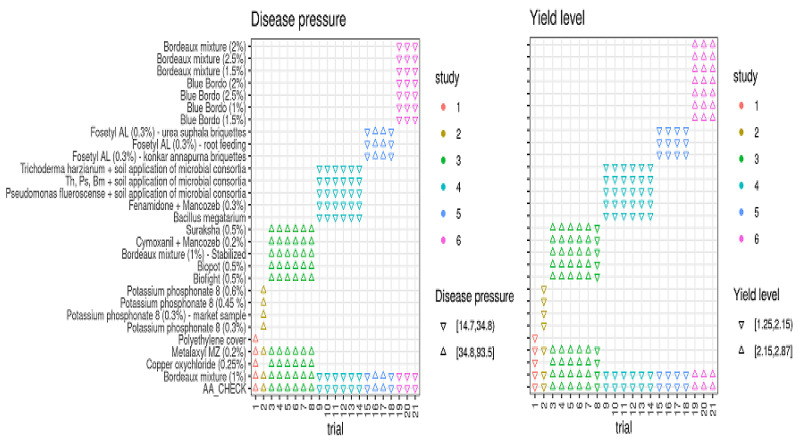
Categorization of trials based on trial mean values for disease pressures and yield levels among the treatments conducted with multi-environment profiles from 2012 to 2019. Th = *Trichoderma harzianum*; Ps = *Pseudomonas fluorescence*; Bm = *Bacillus megatarium*.

**Figure 4 jof-08-00937-f004:**
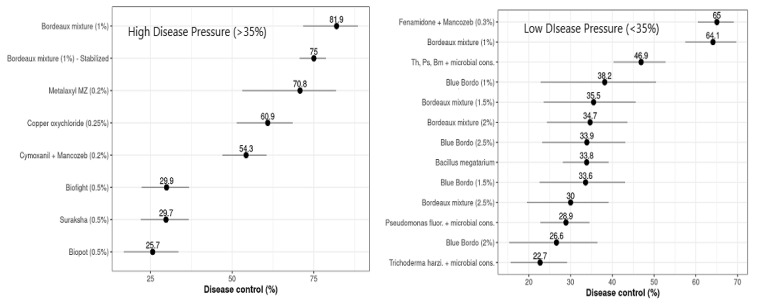
Overall mean control efficiency (%) of evaluated treatments against FRD was estimated by multivariate meta-analysis. Horizontal bars represent 95% confidence intervals; points indicate the differences in control efficiency.

**Figure 5 jof-08-00937-f005:**
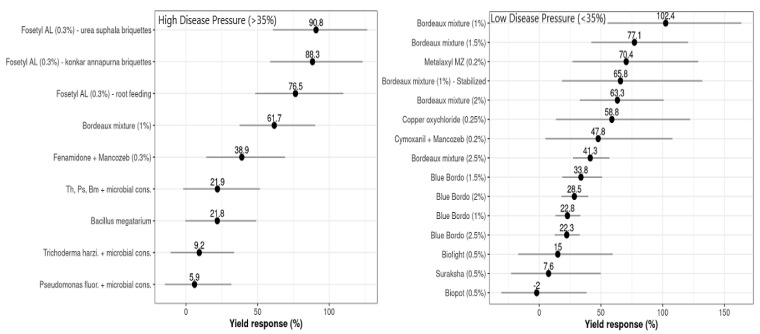
Yield differences (*D*, points) and 95% confidence intervals (horizontal lines) for each evaluated treatment used to control FRD in arecanut at both disease baseline classes (low ≤ 35%, high > 35%). These values were estimated by fitting the network meta-analysis model in R software.

**Table 1 jof-08-00937-t001:** Information about the evaluated treatments against fruit rot disease (FRD) with multi-environment profiles.

Treatments/Fungicides	Dose (%)	Type of Fungicide	Management Principle	Type of Application	Agri-System
Bordeaux Mixture	1.0–5.0	Contact	Chemical	Foliar	Organic
Copper oxychloride	0.20	Contact	Chemical	Foliar	Organic
Metalaxyl + Mancozeb	0.25	Combi	Chemical	Foliar	Inorganic
Potassium Phosphonate	0.3–0.6	Systemic	Chemical	Foliar	Phosphonates
Cymoxanil + Mancozeb	0.30	Combi	Chemical	Foliar	Inorganic
Bordeaux Mixture (Stabilized)	1.00	Contact	Chemical	Foliar	Organic
Fenamidone + Mancozeb	0.30	Combi	Chemical	Foliar	Inorganic
Fosetyl-Al	0.30	Systemic	Chemical	Soil application amended with fertilizers	Phosphonates
Blue Bordo	1.0–5.0	Contact	Chemical	Foliar	Organic
Polyethylene cover	--	--	Mechanical	Bunch Cover	Conventional
Biofight	0.5	Systemic	Bio-product	Foliar	Organic
Biopot	0.5	Systemic	Bio-product	Foliar	Organic
Suraksha	0.5	Systemic	Bio-product	Foliar	Organic
*Bacillus megatarium* (Bm)	200 g/palm	--	Bio-control	Soil application of microbial consortia	Biological
*Trichoderma harzianum* (Th)	200 g/palm	--	Bio-control	Soil application of microbial consortia	Biological
*Pseudomonas fluorescence* (PF)	200 g/palm	--	Bio-control	Soil application of microbial consortia	Biological
Bm + Th + PF consortia	200 g/palm	--	Bio-control	Soil application of microbial consortia	Biological

**Table 2 jof-08-00937-t002:** Descriptions of the locations of the experimental trials conducted in different provinces against FRD in arecanut.

District/Province	Experimental Locations	The Year the Trial Was Conducted
Uttara Kannada (North Canara)	Sirsi	2010
Uttara Kannada (North Canara)	Bilaghi, Siddapura	2013
Shivamogga	Varadamula	2014, 2015, 2016
Shivamogga	Sagara	2014, 2015, 2016
Shivamogga	Tuppooru	2014, 2015, 2016
Shivamogga	Kouti	2014, 2015, 2016
Thirthahalli	Wodeyala	2014, 2015, 2016
Thirthahalli	Bobbi	2014, 2015, 2016
Sagara	Manchale	2015, 2016, 2017
Sagara	Koluru	2015, 2016, 2017
Sagara	Melige	2015, 2016, 2017
Raigad	Shriwardhan	2016
Raigad	Diveagar	2016
Raigad	Nagoli	2016
Raigad	Chaul	2016
Thirthahalli	Agumbe	2016
Thirthahalli	Agumbe	2017
Thirthahalli	Agumbe	2018

**Table 3 jof-08-00937-t003:** Estimated response rate for FRD (*DP*_High_ > 35%) for each treatment compared to the untreated check, as determined via the network meta-analytical model. Percentages of control efficiency and corresponding model statistics are represented.

Treatments ^a^	Effect Size ^c^						Control Efficiency (%) ^d^	
	K ^b^	*L^Inc^*	SE	95% CI	*Z*	P	*C*	95% CI
Intercept	-	4.245	0.054	4.13: 4.35	78.58	0.0001	-	--
Bordeaux mixture (1%)	21	−1.711	0.228	−2.16: −1.26	−7.47	0.0001	81.94	71.72: 88.46
Copper oxychloride (0.25%)	18	−0.938	0.111	−1.15: −0.72	−8.45	0.0001	60.87	51.37: 68.52
Metalaxyl + Mancozeb (0.2%)	21	−1.229	0.241	−1.70: −0.75	−5.09	0.0001	70.76	53.07: 81.78
Biofight (0.5%)	12	−0.355	0.052	−0.45: −0.25	−6.73	0.0001	29.91	22.27: 36.80
Biopot (0.5%)	12	−0.296	0.057	−0.40: −0.18	−5.17	0.0001	25.66	16.83: 33.55
Bordeaux mixture (1%)–Stabilized	16	−0.386	0.082	−1.54: −1.22	−16.87	0.0001	74.99	70.62: 78.71
Cymoxanil + Mancozeb (0.2%)	16	−0.782	0.075	−0.92: −0.63	−10.41	0.0001	54.25	47.00: 60.51
Suraksha (0.5%)	12	−0.353	0.053	−0.45: −0.24	−6.62	0.0001	29.74	

The network meta-analysis model fitted the data collected from the field trials carried out under multi-location conditions in India. ^a^ Active ingredients of treatments used against FRD in arecanut under field conditions. ^b^ The number of trials selected for each specific treatment with untreated controls. ^c^ Mean logarithmic rate ratio (*L^Inc^*) for the average effect of each treatment on FRD compared to the untreated check standard error (SE) of *L^Inc^* and 95% confidence interval (CI) containing *L^Inc^*; *Z* (standard normal) statistic from the network meta-analytical model; *p* = probability value (level of significance). ^d^ Average percentage of control efficiency (*C*) and 95% CI containing *C*; Test for Residual Heterogeneity; QE (df = 52) = 7689.7531, *p*-value < 0.0001; Test of Moderators (coefficients 2:9); QM (df = 8) = 358.1598, *p*-value < 0.0001.

**Table 4 jof-08-00937-t004:** Estimated response rates for FRD (*DP*_Low_ > 35%) for each treatment compared to the untreated check as determined via the network meta-analytical model. Percentages of control efficiency and corresponding model statistics are represented.

Treatments ^a^	Effect Size ^c^						Control Efficiency (%) ^d^	
	K ^b^	*L^Inc^*	SE	95% CI	*Z*	P	*C*	95% CI
Intercept	-	3.126	0.060	3.08: 3.24	52.079	0.0001	-	--
Bordeaux mixture (1%)	21	−1.024	0.086	−1.19: −0.85	−11.830	0.0001	64.10	57.46: 69.70
*Bacillus megatarium* (Bm) + microbial consortium	9	−0.413	0.042	−0.49: −0.33	−9.754	0.0001	33.84	28.11: 39.11
Fenamidone + Mancozeb (0.3%)	9	−1.051	0.062	−1.17: −0.92	−16.804	0.0001	65.05	60.48: 69.07
*Pseudomonas fluorescence* (Ps) + microbial consortium	9	−0.340	0.042	−0.42: −0.25	−8.084	0.0001	28.88	22.75: 34.52
Th, Ps, Bm + microbial consortium	9	−0.632	0.059	−0.74: −0.51	−10.604	0.0001	46.87	40.29: 52.74
*Trichoderma harzianum* (Th) + microbial consortium	9	−0.257	0.044	−0.34: −0.17	−5.771	0.0001	22.69	15.63: 29.61
Blue Bordo (1.5%)	6	−0.409	0.078	−0.56: −0.25	−5.225	0.0001	33.58	22.56: 43.03
Blue Bordo (1%)	6	−0.480	0.044	−0.70: −0.25	−4.249	0.0001	38.17	22.82: 50.46
Blue Bordo (2.5%)	6	−0.413	0.078	−0.56: −0.26	−5.402	0.0001	33.88	23.18: 43.10
Blue Bordo (2%)	6	−0.309	0.113	−0.45: −0.16	−4.226	0.0001	26.61	15.25: 36.42
Bordeaux mixture (1.5%)	9	−0.438	0.086	−0.60: −0.26	−5.053	0.0001	35.50	23.54: 43.59
Bordeaux mixture (2.5%)	9	−0.356	0.071	−0.49: −0.21	−5.016	0.0001	30.00	19.53: 39.10
Bordeaux mixture (2%)	9	−0.425	0.075	−0.57: −0.27	−5.671	0.0001	34.64	24.31: 43.60

The network meta-analysis model fitted the data collected from the field trials carried out under multi-location conditions in India. ^a^ Active ingredients of treatments used against FRD in arecanut under field conditions. ^b^ The number of trials selected for each specific treatment with untreated controls. ^c^ Mean logarithmic rate ratio (*L^Inc^*) for the average effect of each treatment on FRD compared to untreated checks; standard error (SE) of *L^Inc^* and 95% confidence interval (CI) containing *L^Inc^*; *Z* (standard normal) statistic from the network meta-analytical model; *p* = probability value (level of significance). ^d^ Average percentage of control efficiency (*C*) and 95% CI containing *C*; Test for Residual Heterogeneity; QE (df = 59) = 7739.0720, *p*-value < 0.0001; Test of Moderators (coefficients 2:14); QM (df = 9) = 432.0348, *p*-value < 0.0001.

**Table 5 jof-08-00937-t005:** Average arecanut yield differences (*D*) among treated plots and untreated controls (low yield level) estimated through the network meta-analysis with related model parameters and computed yield response (%) for the evaluated treatments for FRD (*DP*_High_ > 35%).

Treatments ^a^	Effect Size ^c^						Yield Response (%) ^d^	
	K ^b^	*D*	SE	95% CI	*Z*	P	*R*	95% CI
Intercept	-	−0.043	0.106	−0.25: 0.16	−0.407	0.6838	-	--
Bordeaux mixture (1%)	10	0.480	0.083	0.31: 0.64	5.781	0.0001	61.70	37.40: 90.30
*Bacillus megatarium* (Bm) + microbial consortium	6	0.197	0.103	−0.05: 0.39	1.913	0.0557	21.80	−0.48: 49.10
Fenamidone + Mancozeb (0.3%)	6	0.328	0.100	0.13: 0.54	3.269	0.0011	38.90	14.10: 69.20
*Pseudomonas fluorescence* (Ps) + microbial consortium	6	0.057	0.111	−0.16: 0.27	0.518	0.6042	5.93	−14.80: 31.70
Th, Ps, Bm + microbial consortium	6	0.198	0.111	−0.01: 0.41	1.782	0.0747	21.90	−1.69: 51.60
*Trichoderma harzianum* (Th) + microbial consortium	6	0.088	0.102	−0.11: 0.28	0.860	0.3897	9.23	−10.70: 33.60
Fosetyl AL (0.3%)–Konkan briquettes	4	0.633	0.087	0.46: 0.80	7.279	0.0001	88.30	58.80: 123.0
Fosetyl AL (0.3%)–root feeding	4	0.567	0.088	0.39: 0.74	6.408	0.0001	76.50	48.30: 110.0
Fosetyl AL (0.3%)–Urea briquettes	4	0.646	0.087	0.47: 0.81	7.370	0.0001	90.80	60.70: 127.0

The network meta-analysis model fitted the data collected from the field trials carried out under multi-location conditions in India. ^a^ Active ingredients of the evaluated treatments for FRD in arecanut. ^b^ The number of trials selected for each specific treatment with untreated controls. ^c^ Mean yield differences (*D*, kg/palm) for each treatment corresponding to untreated checks; standard error (SE) of *D* and 95% confidence interval (CI) around *D*; *Z* (standard normal) statistic from the network meta-analytical model; *p* = probability value (level of significance). ^d^ Average yield response (*R*) computed by back-transformation of estimated FRD incidence and 95% CI (lower and upper limits) containing *R*; Test for Residual Heterogeneity; QE (df = 52) = 63.7404, *p*-value = 0.1274; Test of Moderators (coefficients 2:10); QM (df = 9) = 194.2997, *p*-value < 0.0001.

**Table 6 jof-08-00937-t006:** Average arecanut yield differences (*D*) among treated and untreated controls (high yield levels), as estimated through the network meta-analysis, with related model parameters and computed yield responses (%) for the evaluated treatments for FRD (*DP*_Low_ ≤ 35%).

Treatments ^a^	Effect Size ^c^						Yield Response (%) ^d^	
	K ^b^	*D*	SE	95% CI	*Z*	P	*R*	95% CI
Bordeaux mixture (1%)	8	0.705	0.135	0.44: 0.96	5.222	0.0001	102.0	55.40: 164.0
Copper oxychloride (0.25%)	5	0.462	0.171	0.12: 0.79	2.700	0.0069	58.80	13.50: 122.0
Metalaxyl + Mancozeb (0.2%)	5	0.533	0.150	0.23: 0.82	3.551	0.0004	70.40	27.0: 129.0
Biofight (0.5%)	5	0.140	0.166	−0.18: 0.46	0.839	0.4012	15.00	−17.10: 59.50
Biopot (0.5%)	5	0.020	0.176	−0.36: 0.32	−0.117	0.9062	−2.05	−30.50: 38.30
Bordeaux mixture (1%)–Stabilized	5	0.505	0.171	0.16: 0.84	2.946	0.0032	65.80	18.40: 132.0
Cymoxanil + Mancozeb (0.2%)	5	0.390	0.174	0.04: 0.73	2.240	0.0250	47.80	5.01: 108.0
Suraksha (0.5%)	5	0.073	0.169	−0.25: 0.40	0.432	0.6651	7.60	−22.80: 49.90
Blue Bordo (1.5%)	3	0.291	0.061	0.17: 0.41	4.742	0.0001	33.80	16.60: 50.90
Blue Bordo (1%)	3	0.205	0.041	0.12: 0.28	4.924	0.0001	22.80	13.20: 33.30
Blue Bordo (2.5%)	3	0.201	0.042	0.11: 0.28	4.750	0.0001	22.30	12.50: 32.80
Blue Bordo (2%)	3	0.250	0.043	0.16: 0.33	5.814	0.0001	28.50	18.10: 39.80
Bordeaux mixture (1.5%)	3	0.571	0.112	0.35: 0.79	5.105	0.0001	77.10	42.20: 121.0
Bordeaux mixture (2.5%)	3	0.346	0.053	0.24: 0.44	6.529	0.0001	41.30	27.40: 56.80
Bordeaux mixture (2%)	3	0.490	0.105	0.28: 0.69	4.657	0.0001	63.30	32.90: 101.0

The network meta-analysis model fitted the data collected from the field trials carried out under multi-location conditions in India. ^a^ Active ingredients of the evaluated treatments for FRD in arecanut. ^b^ The number of trials selected for each specific treatment and untreated controls. ^c^ Mean yield difference (*D*, kg/palm) for each treatment corresponding to the untreated check; standard error (SE) of *D* and 95% confidence interval (CI) around *D*; *Z* (standard normal) statistic from the network meta-analytical model; *p* = probability value (level of significance). ^d^ Average yield response (*R*) computed by back-transformation of estimated FRD incidence and 95% CI (lower and upper limits) containing *R*; Test for Residual Heterogeneity; QE (df = 56) = 522.6497, *p*-value < 0.0001; Test of Moderators (coefficients 2:16); QM (df = 15) = 404.3491, *p*-value < 0.0001.

## Data Availability

The data presented in this study are available on request from the corresponding authors.

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
