# Peer review of "Control Efficiency and Yield Response of Chemical and Biological Treatments against Fruit Rot of Arecanut: A Network Meta-Analysis"

_jof, 2022, doi:10.3390/jof8090937_

Round 1

Reviewer 1 Report

In the original article ‘Control Efficiency and Yield Response of Chemical and Biological Treatments Against Fruit Rot of Arecanut: A Network Meta-analysis’, Balanagouda Patil et al performed a quantitative synthesis analysis of the efficacy of fungicides and biological treatments against arecanut FRD.

Suggest to accept in current form but with few minor modifications as mentioned below

Line 24: write “data were” instead of “Data was” gathered from 21 field  

Line 27: write “data were” instead of “data was” evaluated for the estimated effects of disease pressure

Line 181: write “LInc was” instead of LInc (derived after fitting the meta-analytical model) “were”

Line 287: “The calculated LInc values were” instead of “The calculated LInc values was” ranging from -1.711

Lines 47, 151, 277, 447: you don’t have to repeat fruit rot disease (FRD), mention either full name or abbreviation at a time

Lines 476-477: delete the second “was” in the sentence “Similarly, there was statistically significant variation “was” observed among 

Author Response

Author’s Response to Reviewer 1 Comments

Dear reviewer

 Thank you very much, appreciate the time and effort that you dedicated to providing feedback on our manuscript and are grateful for the constructive comments. We have incorporated all of the suggestions and those changes are highlighted (in track changes mode) within the manuscript. Please see below, in red, for a point-to-point response to the reviewer’s comments and concerns.  

Minor Corrections

Point 1: Line 24: write “data were” instead of “Data was” gathered from 21 field  

Response 1: This suggestion was accepted and modified accordingly as revealed in the abstract section

Point 2: Line 27: write “data were” instead of “data was” evaluated for the estimated effects of disease pressure

Response 2: Corrected in the manuscript as per suggestion

Point 3: Line 181: write “LInc was” instead of LInc (derived after fitting the meta-analytical model) “were”

Response 3: The sentence has been changed and modified accordingly.

Point 4: Line 287: “The calculated LInc values were” instead of “The calculated LInc values was” ranging from -1.711

Response 4: The sentence has been modified as per correction

Point 5: Lines 47, 151, 277, 447: you don’t have to repeat fruit rot disease (FRD), mention either full name or abbreviation at a time.

Response 5: Full name has been mentioned when it is used the first time and subsequently the term FRD was used throughout the manuscript.

Point 6: Lines 476-477: delete the second “was” in the sentence “Similarly, there was statistically significant variation “was” observed among 

Response 6: The grammatical error in the sentence has been changed as per the comments.

Reviewer 2 Report

The manuscript presents a quantitative synthesis of several studies conducted in India for management of arecanut FRD. The results are communicated effectively, and overall the paper is straightforward to follow. There are a couple areas where language, including in the methods, can be clarified. This is noted on the attached pdf. At the end of the introduction, line 121 and 122, it is mentioned as part of the objectives:

"ii) determine the profitable economic returns for recommending fungicides under volatile arecanut market prices and incurred costs scenarios." 

This would be highly valuable information, however, it is not included in this version of the manuscript. The authors need to either remove this from the introduction, or include the work detailing the methods/results for this section.

Other than that the suggested edits are relatively few and should be quick to address during revision.

Author Response

Author’s Response to Reviewer 2

Dear reviewer

Thank you very much for your kind support and positive response to our manuscript. We appreciate the time and effort that you have rendered to review our manuscript on “Control Efficiency and Yield Response of Chemical and Biological Treatments Against Fruit Rot of Arecanut: A Network Meta-analysis”.

Minor Corrections

Point 1: The manuscript presents a quantitative synthesis of several studies conducted in India for the management of arecanut FRD. The results are communicated effectively, and overall, the paper is straightforward to follow. There are a couple of areas where language, including in the methods, can be clarified. This is noted on the attached pdf.

Response 1: Thank you very much for the positive comments on our manuscript. The suggested queries were corrected and modified throughout the manuscript.

Point 2: At the end of the introduction, lines 121 and 122, it is mentioned as part of the objectives: "ii) determine the profitable economic returns for recommending fungicides under volatile arecanut market prices and incurred costs scenarios”. This would be highly valuable information; however, it is not included in this version of the manuscript. The authors need to either remove this from the introduction or include the work detailing the methods/results for this section.

 Response 2: Accepted your suggestion and as per the reviewer’s suggestion that sentence has been removed from the introduction.  

Point 3: Other than that, the suggested edits are relatively few and should be quick to address during revision.

Response 3: All the typo errors marked across the manuscript have been corrected as revealed in the revised version.

Point 4: Specify which adhesive was used.

Response 4: Specified the adhesive details (RENTON VA 98055) used in the experiments.

Point 5: Need to explain abbreviations in the figure legend.

Response 5: Abbreviations in the figure legend have been explained in the manuscript.

Point 6: Do you need to include more methods on the Wald statistic you are mentioning? Are you indicating an analysis of inconsistency?

Response 6: Wald statistic was utilized to test the consistency or inconsistency between the experiments or designs including the influence of moderator variables on the experiment/treatments.

Round 2

Reviewer 2 Report

Important revisions still need to be made in order to accurately reflect

the work that was conducted and presented in the current study.

It is with kind words that I say, many of these required edits could have

been prevented if the authors spent more time critically reading their

revised manuscript to ensure consistency throughout. It is beneficial to

revise a manuscript over the course of more than a single day or a day and

a half in order to allow our eyes to view our own work fresh.

Line 225, “mean µ

line 226, is “informal Σ matrix” the correct terminology? this terminology is not

common

Line 225, for consistency you should detail what “Si” represents;

folks that have some familiarity with the normal distribution

or statistical models in general can understand, but then again,

the purpose to clarify is for the benefit of the reader,

some of which may not be as familiar; detailing what

Si represents would further the consistency you have

established when you explain that Yi is the vector of responses…

In your response to the review of the original manuscript version,

you have indicated “Wald statistic was utilized to test the

consistency or inconsistency between the experiments or

designs including the influence of moderator variables on

the experiment/treatments.” Thank you for clarifying this,

but again for the benefit of the reader, this type of

information is standard to include in the materials

and methods. Currently this information is lacking

from the Materials and Methods section.

Line 461-463 “We further, demonstrated the mean and

uncertainty (effect size) of meta-analytic model estimates

in risk analysis to make out decisions while selecting a

profitable and economical management approach.”

The risk analysis section is not included in this manuscript.

Details regarding the profitability/economic value of treatments is missing.

This needs to be either included, or this statement needs to be removed.

Line 466-467 “The analysis of raw data, published

in an annual report and technical bulletins, and

unpublished data cannot compare the treatment means

(multivariate comparison tests) within-study and

between-study, which fails to generate quantitative

estimates of yield response (difference or gain in yield).”

While I think I understand what you are trying to say,

the wording should be improved for greater clarity. I

think you mean to say the analysis of individual experiment

data by itself… To clarify the difference in meaning, a

single-stage meta-analysis can be conducted on the raw data

of multiple experiments to yield the results and comparisons you are

referencing.

Line 469-472 “Comparison of treatment means and yield

response is the key factor and relevant information to make

a decision based on economic benefits and profitability of

treatments under the varied range of price scenarios.”

Here is a further reference to economic benefits and

profitability of treatments. Since this information is

not included in the manuscript, the authors should

either remove the statement, or revise it to reference

future work that can be done along these lines which

was not part of the current study.

Line 493-424 “However, we concluded that three appropriate

timed applications are optimal for better management of the

FRD and to sustain yield response.”

Please revise this statement to clarify to what you are

comparing three applications as being better to? The

experiments included in your work all appeared to have

three applications of each treatment (if this is not accurate,

the methods section needs to be updated). Are you comparing

your results based on three applications to a different

number of treatment applications that were used in

citations 21 and 23? I am confused and interested to

have a better understanding of your statement, and

I think this would help other readers as well.

Line 511 references the results of the study as a guide for

assessing... “economic benefit returns on fungicidal investment”

Please include this information or delete this part of the statement.

Line 515-516 “economic benefit and returns”

revise appropriately

Conclusions section

Line 525-526 “In this investigation, fungicides...significantly

increased arecanut yield and profitability…”

The profitability section of the manuscript is not included.

Revise statement appropriately to remove reference to profitability.

Line 530-531 “...we demonstrated that one application of

copper-based fungicides as a curative measure and one

application of systemic oomycete-specific fungicides

could be reliable approaches to reducing…”

The information regarding the data that was included

in the analysis used in this manuscript appear to all

have used three applications of each treatment. Accordingly,

this statement regarding “one” application of either treatment

is not supported. This needs to be revised to be appropriate for

the work presented.

Line 532 “To ensure the probable positive rate of

investment (ROI),…”

While it is okay and even appropriate to comment on

disease management programs that rotate between

different active ingredients/products, greater attention

to this would first need to be included in the Discussion,

since it is not part of the direct results of this study. To

do this, you can synthesize the results of other studies

in the Discussion comparing such programs that include

product alternation to management programs such as the

ones examined in your analysis that do not rotate among

different treatments.

Author Response

Author’s Response to Reviewer 2

Dear reviewer

Thank you very much and we appreciate the time and effort that you dedicated to providing feedback on our manuscript and are grateful for the insightful, constructive comments and valuable improvements to our paper. We have incorporated all of the suggestions and those changes are highlighted (in track changes mode) within the manuscript. Please see below, in red, for a point-to-point response to the reviewer’s comments and concerns. 

Specific comments

Point 1: Important revisions still need to be made in order to accurately reflect the work that was conducted and presented in the current study. It is with kind words that I say, many of these required edits could have been prevented if the authors spent more time critically reading their revised manuscript to ensure consistency throughout. It is beneficial to revise a manuscript over the course of more than a single day or a day and a half in order to allow our eyes to view our own work fresh.

Response 1: Thank you very much for the constructive and critical concerns about our manuscript and as per the suggestions we have modified the manuscript for the benefit of readers as well as clarity to the work presented in the current study.

Point 2: Line 225, “mean µ”; line 226, is “informal Σ matrix” the correct terminology? this terminology is not common

Response 2: The symbolize of µ and Σ are commonly used mathematical symbols in the meta-analytical model, particularly in R METAFOR package. We encourage that to keep it as it is for better understanding.

Point 3: Line 225, for consistency you should detail what “Si” represents; folks that have some familiarity with the normal distribution or statistical models, in general, can understand, but then again, the purpose to clarify is for the benefit of the reader, some of which may not be as familiar; detailing what Si represents would further the consistency you have established when you explain that Yi is the vector of responses…

Response 3: Si represents a variance-covariance matrix of random effects as revealed in the revised manuscript.

Point 4: In your response to the review of the original manuscript version, you have indicated “Wald statistic was utilized to test the consistency or inconsistency between the experiments or designs including the influence of moderator variables on the experiment/treatments.” Thank you for clarifying this, but again for the benefit of the reader, this type of information is standard to include in the materials and methods. Currently, this information is lacking from the Materials and Methods section.

Response 4: The description of Wald statistic has been incorporated in M & M section as per the suggestion.

Point 5: Line 461-463 “We further, demonstrated the mean and uncertainty (effect size) of meta-analytic model estimates in risk analysis to make out decisions while selecting a profitable and economical management approach.” The risk analysis section is not included in this manuscript. Details regarding the profitability/economic value of treatments are missing. This needs to be either included, or this statement needs to be removed.

Response 5: The risk analysis and profitability were used on the general note and as suggested it has been deleted from the sentence.

Point 6: Line 466-467 “The analysis of raw data, published in an annual report and technical bulletins, and unpublished data cannot compare the treatment means (multivariate comparison tests) within-study and between-study, which fails to generate quantitative estimates of yield response (difference or gain in yield).” While I think I understand what you are trying to say, the wording should be improved for greater clarity. I think you mean to say the analysis of individual experiment data by itself… To clarify the difference in meaning, a single-stage meta-analysis can be conducted on the raw data of multiple experiments to yield the results and comparisons you are referencing.

Response 6: The sentence has been corrected according to the suggestion.

Point 7: Line 469-472 “Comparison of treatment means and yield response is the key factor and relevant information to make a decision based on economic benefits and profitability of treatments under the varied range of price scenarios.” Here is a further reference to economic benefits and profitability of treatments. Since this information is not included in the manuscript, the authors should either remove the statement, or revise it to reference future work that can be done along these lines which was not part of the current study.

Response 7: The mention of economic benefits and profitability has been removed from the manuscript. It has been mentioned in the conclusion section as future work.

Point 8: Line 493-424 “However, we concluded that three appropriate timed applications are optimal for better management of the FRD and to sustain yield response.” Please revise this statement to clarify to what you are comparing three applications as being better to? The experiments included in your work all appeared to have three applications of each treatment (if this is not accurate, the methods section needs to be updated). Are you comparing your results based on three applications to a different number of treatment applications that were used in citations 21 and 23? I am confused and interested to have a better understanding of your statement, and I think this would help other readers as well.

Response 8: We hope that the sentence is straightforward and understandable. For effective control of FRD under field conditions its necessary to take up three applications at the appropriate time. All the experiments included in the present work have compared three applications and three spraying in each treatment conducted in 21 trials.

Point 9: Line 511 references the results of the study as a guide for assessing... “economic benefit returns on fungicidal investment” Please include this information or delete this part of the statement. Line 515-516 “economic benefit and returns” revise appropriately

Response 9: The term economic benefits and returns have been removed and the sentence has been revised appropriately.

Point 10: Conclusions section Line 525-526 “In this investigation, fungicides...significantly increased arecanut yield and profitability…” The profitability section of the manuscript is not included Revise statement appropriately to remove reference to profitability.

Response 10: The term profitability has been removed from the conclusion and it has been modified accordingly.

Point 11: Line 530-531 “...we demonstrated that one application of copper-based fungicides as a curative measure and one application of systemic oomycete-specific fungicides could be reliable approaches to reducing…” The information regarding the data that was included in the analysis used in this manuscript appear to all have used three applications of each treatment. Accordingly, this statement regarding “one” application of either treatment is not supported. This needs to be revised to be appropriate for the work presented.

Response 11: The sentence has been revised as per the suggestion

Point 12: Line 532 “To ensure the probable positive rate of investment (ROI),…” While it is okay and even appropriate to comment on disease management programs that rotate between different active ingredients/products, greater attention to this would first need to be included in the Discussion, since it is not part of the direct results of this study. To do this, you can synthesize the results of other studies in the Discussion comparing such programs that include product alternation to management programs such as the ones examined in your analysis that do not rotate among different treatments.

Response 12: The term rate of investment (ROI) used in the conclusion part is on a general note, however, it has been revised in the conclusion section.
